# Paper-Based Microfluidic Analytical Device Patterned by Label Printer for Point-of-Care Blood Glucose and Hematocrit Detection Using 3D-Printed Smartphone Cassette

**DOI:** 10.3390/s24154792

**Published:** 2024-07-24

**Authors:** Zong-Xiao Cai, Ming-Zhang Jiang, Ya-Ju Chuang, Ju-Nan Kuo

**Affiliations:** 1Department of Automation Engineering, National Formosa University, No. 64, Wenhua Rd., Huwei 63201, Yunlin, Taiwan; 11057105@gm.nfu.edu.tw (Z.-X.C.); 40727105@gm.nfu.edu.tw (M.-Z.J.); 2Department of Laboratory Medicine, National Taiwan University Hospital Yunlin Branch, No. 579, Sec. 2, Yunlin Rd., Douliu 640203, Yunlin, Taiwan; y01054@ms1.ylh.gov.tw

**Keywords:** glucose, hematocrit, label printing, microfluidic, paper-based device, point-of-care, smartphone

## Abstract

This study presents a portable, low-cost, point-of-care (POC) system for the simultaneous detection of blood glucose and hematocrit. The system consists of a disposable origami microfluidic paper-based analytical device (μPAD) for plasma separation, filtration, and reaction functions and a 3D-printed cassette for hematocrit and blood glucose detection using a smartphone. The origami μPAD is patterned using a cost-effective label printing technique instead of the conventional wax printing method. The 3D-printed cassette incorporates an array of LED lights, which mitigates the effects of intensity variations in the ambient light and hence improves the accuracy of the blood glucose and hematocrit concentration measurements. The hematocrit concentration is determined quantitatively by measuring the distance of plasma wicking along the upper layer of the origami μPAD, which is pretreated with sodium chloride and Tween 20 to induce dehydration and aggregation of the red blood cells. The filtered plasma also penetrates to the lower layer of the origami μPAD, where it reacts with embedded colorimetric assay reagents to produce a yellowish-brown complex. A color image of the reaction complex is captured using a smartphone inserted into the 3D-printed cassette. The image is analyzed using self-written RGB software to quantify the blood glucose concentration. The calibration results indicate that the proposed detection platform provides an accurate assessment of the blood glucose level over the range of 45–630 mg/dL (R^2^ = 0.9958). The practical feasibility of the proposed platform is demonstrated by measuring the blood glucose and hematocrit concentrations in 13 human whole blood samples. Taking the measurements obtained from commercial glucose and hematocrit meters as a benchmark, the proposed system has a differential of no more than 6.4% for blood glucose detection and 9.1% for hematocrit detection. Overall, the results confirm that the proposed μPAD is a promising solution for cost-effective and reliable POC health monitoring.

## 1. Introduction

Diabetes is a systemic metabolic disease closely associated with insulin secretion in the body. When the body lacks insulin, it cannot deliver glucose from the blood to the cells, causing blood glucose levels to rise from the normal fasting concentration of 70–100 mg/dL. High blood glucose levels have many adverse effects on human health, including kidney damage, nerve damage, impaired vision, and skin conditions. Thus, the American Diabetes Association suggests that glucose levels in diabetics should be strictly controlled to less than 180 mg/dL [1]. Hematocrit is a measure of the proportion of red blood cells in a given volume of whole blood and is often lower in people with anemia and higher in people with cardiovascular disease. In addition, the hematocrit is generally lower in individuals with higher blood glucose levels and higher in those with lower blood glucose levels. According to WHO data, diabetes and kidney disease caused by diabetes killed an estimated 2 million people in 2019, making it one of the major causes of death. Approximately 6% of people worldwide live with diabetes, and its prevalence is rapidly rising in low- and middle-income nations [2]. Consequently, there is an urgent need for low-cost yet effective and accurate methods for monitoring and managing blood glucose and hematocrit levels [3,4], particularly in resource-limited settings. Such methods are crucial for preventing the complications associated with diabetes and hyperglycemia [5,6,7].

Point-of-care testing (POCT) and on-site analysis have attracted significant interest in recent years for applications in medical diagnostics, food safety, and environmental monitoring [8,9,10]. Microfluidic paper-based analytical devices (μPADs), one of the most common types of POCT devices, have many advantages, including affordability, simplicity, good robustness, an equipment-free nature, and user-friendliness [11,12,13]. Furthermore, paper is inexpensive, accessible, biodegradable, and easy to manufacture. Consequently, paper-based μPADs are highly attractive for the fabrication of disposable and portable detection devices, particularly those intended for use in resource-poor nations [14]. The μPAD concept was proposed by Whitesides et al. [15] in 2007. Since then, μPADs have been extensively developed and deployed in a wide variety of fields, including clinical diagnostics [16,17], food safety [18,19], environmental monitoring [20], and COVID-19 detection [21]. As the power and capabilities of smartphones continue to increase, interest in their potential use as portable and convenient POCT detectors has grown [22,23]. Modern smartphones have significant computational power, reasonable memory, and high-resolution cameras capable of quantifying color and illumination intensities [24,25]. Thus, the integration of μPADs with smartphones has emerged as a powerful new paradigm for intelligent on-site analysis [26].

Paper is naturally hydrophilic. Thus, to limit the fluid flow to a particular location or direction, hydrophobic barriers are required. Several techniques are available for producing these barriers, including photolithography [27,28,29], wax printing [30,31], plasma treatment [32,33], and laser treatment [34,35]. Photolithography provides the ability to pattern μPADs with extremely high resolution. However, the photolithography process requires the use of sophisticated and expensive equipment. In addition, the photoresist material used to define the required μPAD configuration is not only expensive but also reduces the mechanical flexibility of the paper. Wax printing offers high speed, ease of use, and high resolution. However, commercial wax printers are unsuitable for batch production due to their high running costs and the low melting point of wax, which can result in distortion of the printed patterns at high temperatures. While plasma treatment offers a straightforward approach for generating hydrophobic patterns on paper without affecting the surface topography or flexibility, it is poorly suited to mass production. Finally, laser treatment can produce high-resolution μPADs, but folding and storing the paper chips can be challenging since the laser process make the substrate brittle and thus susceptible to cracking during folding. For applications such as glucose detection, the ability to mass-produce μPADs is crucial because of the sheer scale of diabetes and related diseases worldwide. Thus, while the aforementioned methods have undeniable benefits in performing small-scale detection tasks, they are less suited to glucose detection for diabetes on a larger scale. Consequently, more effective and scalable methods for the mass production of μPADs are still required.

Many methods can be applied for the μPAD detection of analytes, including colorimetry [36,37,38], electrochemistry [39,40], mass spectrometry [41], chemiluminescence [42,43], and fluorescence [44,45]. However, electrical detection methods require the use of power supplies and delicate circuitry to measure the current or impedance. Chemiluminescence or fluorescence approaches need an image sensor interfaced with an expensive fluorescence instrument and are thus poorly suited for POCT applications. Mass spectrometry methods also require a costly and bulky apparatus. Among these detection methods, colorimetric methods offer several key advantages, including low cost, minimal equipment requirements, easy operation, simple signal output, and good versatility. Pregnancy and urine test strips are the most common colorimetric detection devices used on-site. However, these test strips provide only a semi-qualitative or yes/no detection capability. Thus, there is still a need for more sophisticated and quantitative detection devices for POCT applications that require more sensitive and accurate measurement [24]. In the case of whole human blood samples, the natural red background color of hemoglobin complicates the identification of specific analytes using colorimetric methods. Thus, μPAD devices designed for diabetes monitoring should ideally provide the ability to separate the plasma from the whole blood sample to enhance the accuracy of the analytical results [46].

To address these requirements, this study proposes a portable, low-cost, point-of-care (POC) system for the simultaneous detection of blood glucose and hematocrit in whole human blood samples. The system comprises an origami μPAD fabricated using a commercial label printing machine and a 3D-printed optical cassette to facilitate detection using a commercial smartphone. The origami μPAD contains a single hematocrit test layer and four plasma filtering layers to improve the color uniformity and intensity of the blood plasma, thereby enhancing the detection accuracy. The hematocrit test layer is pretreated with sodium chloride (NaCl) and Tween 20 solutions, which prompt the dehydration of the red blood cells (RBCs) and suppress their flowability in the hematocrit test layer. Thus, the plasma flows ahead of the RBCs and is effectively separated from the whole blood sample. As the plasma flows along the hematocrit test layer, it also penetrates through the chip to the lower layer, where it reacts with embedded colorimetric assay reagents to produce a yellowish-brown complex. Once the reaction is complete, the paper chip is inserted into the optical cassette and observed using the camera of a commercial smartphone. The hematocrit content is quantitatively determined by measuring the plasma wicking distance along the upper layer of the origami μPAD. Additionally, the glucose concentration is derived by analyzing the RGB intensity of the reaction complex using self-written RGB program installed on the phone as an app. The accuracy of the blood glucose measurements is enhanced by illuminating the detection region of the paper chip with LED lights built into the 3D-printed cassette to minimize the effects of ambient lighting variations.

The measurement results obtained for the blood glucose and hematocrit concentrations in 13 human whole blood samples are shown to be in good agreement with those obtained using commercial meters. Thus, overall, the platform provides a competitive technology for POCT and clinical applications, particularly in resource-poor areas.

## 2. Materials and Methods

### 2.1. Design of Origami μPAD

Figure 1 shows the configuration and dimensions of the origami μPAD, consisting of a single hematocrit test layer and four folded layers for plasma separation, filtration, and colorimetric reaction. The sample area of the hematocrit test layer is coated with a NaCl/Tween 20 solution. When the whole blood sample is dripped onto the sample area, some of the sample wicks along the hematocrit test layer, while the remainder penetrates through the stacked layers towards the blood glucose test zone in the lowest layer of the paper chip. As the sample flows along the hematocrit test layer, the RBCs dehydrate under the effects of the NaCl/Tween 20 reagent. As a result, their movement is impeded, such that only the plasma wicks along the test strip. Layers 2 and 3 of the origami μPAD, located directly beneath the whole blood sample area when the chip is folded, are also pretreated with NaCl and Tween 20 to filter out the RBCs. Meanwhile, layers 4 and 5 are coated with colorimetric assay reagents, such that a yellowish-brown complex is formed when the filtered plasma reaches the blood glucose test zone. Once the colorimetric reaction process is complete, the μPAD is transferred to the optical cassette, and the hematocrit and glucose concentrations are determined using self-written apps installed on the smartphone.

### 2.2. Fabrication of Origami μPAD

The origami μPADs were fabricated using a label printing technique, as shown in Figure 2. The μPAD layout was designed using label printer editing software (BarTender 2021 UltraLite Edition, Seagull Scientific, Washington, DC, USA) (Figure 2a), with multiple μPADs placed side by side on a single page to facilitate mass production. The μPADs were printed using a commercial label printer (TTP-345, TSC Printronix Auto ID, New Taipei City, Taiwan) that transferred a thermal transfer ribbon (EG-18, DNP, Tokyo, Japan) to the surface of a piece of filter paper (Advantec No. 1, ADVANTEC, Kashiwa, Japan) (Figure 2b). After printing, the filter paper was heated on a hotplate at 170 °C for 15 min to allow the ribbon to permeate through the paper thickness, thereby creating hydrophobic boundaries that replicated the μPAD design (Figure 2c). Finally, the filter paper was cut into individual paper-based chips for experimental testing (Figure 2d).

The baking temperature used in the present method was higher than the typical wax-printing temperature since the toner particles in the transfer ribbon required a higher temperature to melt and impregnate the cellulose microfibers of the paper, thereby avoiding distortion of the printed patterns at high temperatures. Moreover, label printers are better suited for batch production and have lower operating costs than wax printers. Thus, overall, the label printing technique offers a simple, efficient, and cost-effective method for the mass production of origami μPADs with high precision and reproducibility.

### 2.3. Smartphone-Based Optical Cassette and Self-Written Apps

Smartphones provide a low-cost and effective technique for μPAD-based colorimetric detection [47,48]. However, when using smartphones to determine the color intensity of a reaction complex, variations in the ambient light conditions may affect the detection performance. In the present study, this problem was addressed using the optical cassette shown in Figure 3, consisting of a smartphone holder, an optical chamber, and a μPAD holder. The cassette was fabricated using a 3D printer (Original, Snapmaker, Shenzhen, China) and was fitted with six LED lights with a 5500 K color temperature to enable accurate and precise color intensity detection.

An affordable smartphone (HTC One M8, HTC Corp., New Taipei City, Taiwan) was used for detection purposes. The smartphone emulator Android Studio (Dolphin version) was used to develop two applications to detect the hematocrit and glucose concentrations in the whole blood sample, respectively. The apps were designed to perform various operations, including image capture, image processing, distance measurement (hematocrit concentration), RGB intensity reading and processing (glucose concentration), quantitative computation, and test result output. The user interfaces of the two applications were designed to be simple and intuitive, and the algorithms used to process the images were optimized for both precision and speed. In addition, the two apps provide precise control of exposure time, light sensitivity, and white balance, as well as analytical calibration capabilities.

### 2.4. Reagent Preparation and Glucose Colorimetric Assay

Animal blood (sheep) was obtained from Taiwan Prepared Media (TPM, Taichung City, Taiwan). Potassium iodide (KI) was purchased from Cheng E Chemical Engineering (Taipei, Taiwan). Sodium chloride (NaCl) was sourced from Shimakyu (Osaka, Japan). Glucose anhydrous was purchased from Scharlau (Barcelona, Spain). Glucose oxidase (GOx), horseradish peroxidase (HRP), and Tween 20 were acquired from Sigma-Aldrich (Merck, Taipei, Taiwan). Phosphate-buffered saline (PBS) was used as the solvent. All the enzyme and coenzyme solutions were freshly prepared and used without intermediate storage. Moreover, all the detection experiments were performed at room temperature (25 °C).

For the glucose colorimetric assay, a potassium iodide (KI) color indicator was used to verify the linear range of the proposed platform. In accordance with ref. [49], the assay was performed using a solution of 0.6 M KI, 1 mg of HRP mixed with 5 mL of PBS, and 4.3 mg of GOx mixed with 10 mL of PBS. In the assay process, the interaction between the glucose in the blood sample and the GOx produced hydrogen peroxide (H_2_O_2_) and gluconic acid. The H_2_O_2_ was then reduced to H_2_O by the catalytic reaction of HRP, and the color indicator (KI) was oxidized, resulting in the formation of a complex with a yellowish-brown color [50].

### 2.5. Experimental Process

Figure 4 shows the main steps in the experimental detection procedure. Prior to the detection experiments, reagents were coated on the reaction zones of the μPAD and allowed to diffuse and dry at room temperature (25 °C) for 10 min (Figure 4a). Reaction zone 1 was coated with 5 μL of NaCl/Tween 20 solution to facilitate hematocrit detection, while reaction zones 2 and 3 were coated with 1 μL of the same mixture to facilitate glucose detection. In addition, 1 μL of 0.6 M KI and glucose reagent was applied to both reaction zone 4 and reaction zone 5 to facilitate the glucose assay (Figure 4a). After the reagents were dry, the μPAD was folded (Figure 4b), and 5 μL of blood sample was dripped onto reaction zone 1 (Figure 4c). The μPAD was left to stand for 2 min to allow for plasma separation and colorimetric reaction. The μPAD was then unfolded (Figure 4d) and inserted into the optical cassette, where it was observed using the smartphone camera (Figure 4e). Finally, the blood glucose concentration and hematocrit value were determined using the self-written apps installed on the phone (Figure 4f).

## 3. Results and Discussion

### 3.1. Characterization of Label-Printed μPAD

As described in Section 2.2, the hydrophobic barriers in the μPAD were patterned using ribbon ink and a commercial label printer (TTP-345, TSC Printronix Auto ID, New Taipei City, Taiwan). With a resolution of over 300 dots per inch (dpi), the printer ensured excellent precision and repeatability of the produced μPADs by uniformly distributing the ribbon ink on the surface of the filter paper. Figure 5a–d show the results obtained from hydrophobicity tests performed using blue dye following various treatments of the filter paper. Figure 5a,b show the surfaces of the filter paper before and after printing, respectively. In the former case, when blue ink was dropped on the reaction zone of the filter paper, it seeped beyond the boundaries of the reaction area owing to the lack of hydrophobic barriers (Figure 5a). During the printing process, the filter paper passed through a fusion unit inside the label printer, which partially melted the toner microparticles and bound them to the surface. However, the short heating time was insufficient to impregnate the hydrophobic ribbon ink through the thickness of the paper substrate. Thus, the blue dye again seeped beyond the boundaries of the reaction region (Figure 5b). Figure 5a,b also present SEM images (200× magnification) of the filter paper morphology before and after printing, respectively. The non-printed paper had a porous fiber structure, as shown in Figure 5a. It was thus naturally hydrophilic and allowed the dye to flow easily through the paper structure under the effects of capillary action. During the printing process, ribbon ink was applied to the surface of the filter paper. However, no significant change in the paper morphology occurred (Figure 5b). Hence, the blue dye was again able to seep beyond the printed boundaries.

Figure 5c shows the surface morphology of the label-printed filter paper following heating at 170 °C for 15 min. In this case, the melted toner and wax penetrated deeply into the three-dimensional porous fiber structure of the filter paper, thus forming robust hydrophobic barriers that prevented capillary-induced diffusion of the blue ink out of the reaction region. In other words, the label printing technique successfully overcame the main limitation of wax-based patterning, namely, the difficulty of accurately controlling the flow of melted wax into the porous capillaries of the paper to precisely define the required hydrophobic barriers [30]. For the label printer used in the present study, the toner consisted mainly of styrene-acrylate resin, which softens at temperatures in the range of 100–150 °C [51]. Consequently, when heated to 170 °C, the toner transforms into a viscous fluid that progressively wicks into the paper within a narrowly defined area [52]. Due to the high softening point of the resin (greater than 100 °C), the use of ribbon ink rather than wax to create the hydrophobic boundaries has the further advantage that the resulting devices are highly thermally stable.

Figure 5d shows the morphology of the filter paper after treatment with NaCl and Tween 20. The reagent solution forms a crystalline substance attached to the paper fibers. When the blood sample is dripped onto the reaction region of the μPAD, the water in the blood dissolves the crystals, which creates osmotic pressure and leads to the dehydration of the RBCs under the effects of the NaCl. Meanwhile, the Tween 20 acts as a surfactant, which reduces the surface tension between the liquid and solid components of the sample and allows the plasma to penetrate the fiber structure of the filter paper fiber. Hence, a separation of the RBCs and blood plasma occurs.

### 3.2. Optimization of Plasma Separation Effect

In the proposed μPAD, the hematocrit and glucose detection performance is fundamentally dependent on the efficiency of the plasma separation process. Therefore, a preliminary investigation was conducted using animal (sheep) blood to determine the optimal composition of the NaCl/Tween 20 reagent coated on the first three reaction layers of the μPAD (Figure 4a). As shown in Figure 6, six reagent solutions with varying concentrations of NaCl and Tween 20 were prepared. For each mixture, 5 μL of reagent solution was applied to reaction zone 1, while 1 μL was applied to reaction zone 2 and reaction zone 3.

For a reagent consisting solely of 4% NaCl (diluted in DI water), only limited plasma separation occurred, and reaction layers 4 and 5 of the paper chip remained dry (Figure 6a). When 0.25% and 0.5% Tween 20 were added to the reagent, the plasma reached reaction layers 4 and 5 but still contained RBCs, as shown in Figure 6b,c, respectively. The NaCl concentration was thus increased to 10%. However, a small number of RBCs still reached reaction layer 5 (Figure 6d). Consequently, the Tween 20 content was increased to 0.25%. In this case, the plasma reached reaction layer 5, with no visible trace of RBCs (Figure 6e). However, when the Tween 20 content was further increased to 0.5%, the plasma failed to penetrate layer 5 due to the interaction with the high NaCl concentration (Figure 6f). Thus, 10% NaCl + 0.25% Tween 20 was chosen as the optimal reagent composition for the subsequent experiments.

### 3.3. Hematocrit Calibration Curve

The hematocrit detection performance of the origami μPAD was evaluated using seven control samples with known hematocrit values. Briefly, the hematocrit concentration of the original sheep blood sample was determined by centrifugation. Control samples with hematocrit concentrations spanning the clinically relevant range of human hematocrit (25–55%) were prepared at 5% intervals by removing hematocrit from the original sample as required. For each control sample, 5 μL of sheep blood was dripped onto the reaction zone on the upper layer of the μPAD, and the wicking distance was measured. The measurement results are presented in Figure 7a,b. As shown in Figure 7a, the wicking distance decreased as the hematocrit concentration increased. According to the calibration curve in Figure 7b, based on five independent measurements for each sample (each data point is a mean ± standard deviation of n = 5 assays), the wicking distance (Y) varied linearly with the hematocrit concentration (X) over the considered range of 25–55% as Y = −0.089675X + 8.009509, with a correlation coefficient of R^2^ = 0.9741. The correlation coefficient is close to the ideal value of 1, indicating that the calibration equation has good linearity and reliability. Thus, it was implemented in the smartphone application to determine the hematocrit of blood samples with unknown hematocrit concentrations.

### 3.4. Blood Glucose Calibration Curve

The glucose determination performance of the μPAD system was investigated by diluting sheep plasma samples with PBS to obtain control samples with glucose concentrations ranging from 45 to 630 mg/dL, covering the full clinical range associated with normal and diabetic glucose levels. For each sample, the R, G, and B intensity values were independently measured at least five times to ensure the reliability of the results. An analysis of the measured R, G, and B values indicated that the intensity ratio of G/(R + G + B) provided the best fit with the glucose concentration of the control samples. As shown in Figure 8, the optimal calibration equation was determined to be Y = −0.000167 X + 0.377428, with a near-ideal correlation coefficient of R^2^ = 0.9958. The μPAD system has a detection sensitivity of 0.000167 (a.u.)/(mg/dL), indicating an ideal inverse proportional relationship. Thus, the formula was implemented in the second app on the smartphone to facilitate the glucose determination of unseen blood samples. The limit of detection for glucose detection was calculated to be around 4 mg/dL using the mean (blank) + 3SD (standard deviation) [53], which is comparable to previously reported colorimetric methods (1.8–19.8 mg/dL) [21,54].

### 3.5. Practical Application of μPAD System to Hematocrit and Glucose Determination in Human Whole Blood Samples

The practical applicability of the proposed origami μPAD system was investigated by measuring the hematocrit and glucose levels in whole blood samples collected from 13 adult volunteer donors through fingertip blood collection. Figure 9a shows a typical result obtained for the glucose concentration in one of the samples. It can be seen that the detection result (131.1 mg/dL) is in good agreement with that obtained using a commercial glucose meter (133 mg/dL, ACCU-CHEK Guide, Roche, Basel, Switzerland). Figure 9b shows the detection result for the hematocrit concentration in one of the whole blood samples. The determination result (39.6%) is consistent with that obtained using a commercial hematocrit meter (43%, MD6, FORA, Taipei, Taiwan).

Table 1 compares the blood glucose and hematocrit measurements obtained by the μPAD and smartphone system with those obtained by the two commercial meters for each of the 13 whole blood samples. Taking the results of the two commercial meters as a benchmark, the relative error is calculated by comparing the measured value of the μPAD to the results of the two commercial meters. The deviation between the two sets of results is less than 6.4% (ranging from 1.4 to 6.4%) for the glucose measurements and 9.1% (ranging from 1.9 to 9.1%) for the hematocrit measurements. Thus, the basic feasibility of the proposed platform for practical applications is confirmed. Notably, the μPAD system not only rivals the performance of the two commercial meters but also does so at a far lower cost. In particular, each μPAD costs just USD 0.01 (lab-fabricated; labor not included), while the optical cassette with built-in LED lights costs less than USD 10. In contrast, the commercial glucose and hematocrit meters used in this study have approximate costs of USD 100 and USD 80, respectively. Commercial glucose and hematocrit test strips cost around USD 0.5 and USD 2, respectively.

## 4. Conclusions

This study has presented an integrated platform for performing the simultaneous quantitative detection of glucose and hematocrit in human whole blood samples. The platform comprises an origami μPAD, a 3D-printed smartphone-based cassette, and a smartphone. The reaction zones of the μPAD are pretreated with a solution of 10% NaCl and 0.25% Tween 20 to optimize the plasma separation and filtration effect. In addition, the detection zone of the μPAD is coated with a solution of 0.6 M KI, 1 mg of HRP mixed with 5 mL of PBS, and 4.3 mg of GOx mixed with 10 mL of PBS to facilitate a colorimetric reaction. In the detection process, 5 μL of human whole blood is dropped on the reaction zone of the upper layer of the μPAD and is subsequently filtered under the effect of the NaCl/Tween 20 reagent. Part of the separated plasma wicks along the upper layer of the paper chip, while the remainder penetrates through the thickness of the chip to the reaction zone, where it undergoes a colorimetric reaction and produces a complex with a yellowish-brown color. Following the reaction process, the μPAD is transferred to the optical cassette and observed by the camera of a commercial smartphone (HTC One M8, HTC Corp., New Taipei City, Taiwan). The hematocrit and glucose concentrations are then derived using self-written applications installed on the phone and designed to determine the wicking distance of the plasma along the upper layer of the chip (hematocrit detection) and the G/(R + G + B) intensity ratio of the reaction complex (glucose concentration).

The origami μPADs are printed using a commercial label printer. This not only results in more precise hydrophobic barriers than those created using a conventional wax-printing method but also enables mass production at a significantly lower cost. In addition, the optical cassette is equipped with LED lights, which mitigate the effects of variations in the ambient light and therefore increase the accuracy of the colorimetric detection results. The experimental results show that the hematocrit and glucose concentration measurements obtained for 13 real-world human whole blood samples deviate by no more than 9.1% and 6.4%, respectively, from the measurements obtained using commercial hematocrit and glucose meters. Overall, the results indicate that the proposed platform provides a viable, low-cost solution for POC hematocrit and glucose determination in whole blood samples. It thus offers a promising solution for POC glucose monitoring and management, particularly in resource-poor settings.

## Figures and Tables

**Figure 1 sensors-24-04792-f001:**
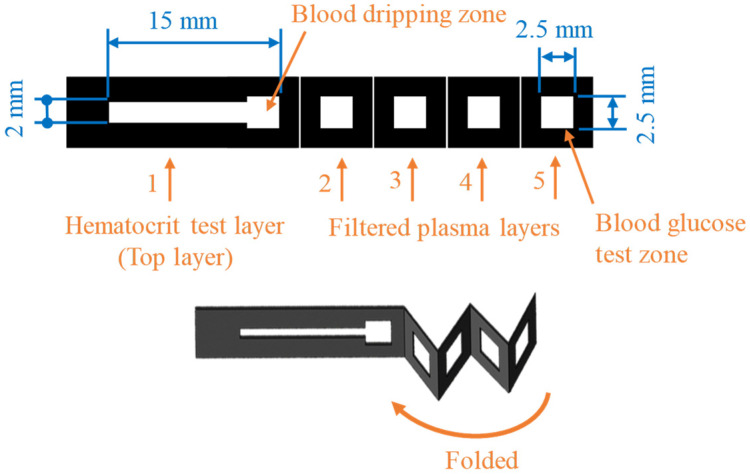
Schematic illustration of origami μPAD.

**Figure 2 sensors-24-04792-f002:**
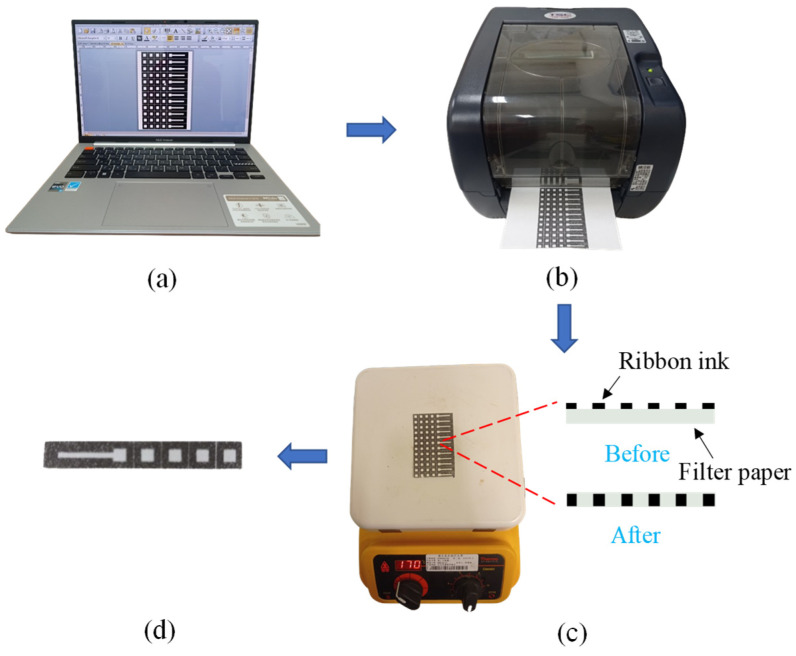
Origami μPAD fabrication process: (**a**) design, (**b**) label printing, (**c**) baking, and (**d**) cutting.

**Figure 3 sensors-24-04792-f003:**
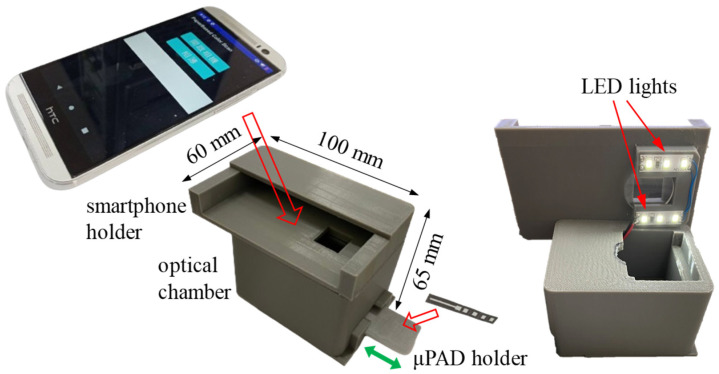
Images and dimensions of smartphone-based optical cassette incorporating smartphone holder, optical chamber, μPAD holder, and LED lights.

**Figure 4 sensors-24-04792-f004:**
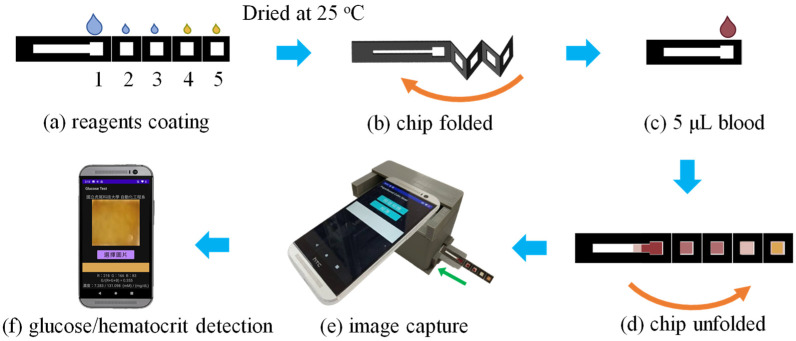
Operating steps of experimental procedure.

**Figure 5 sensors-24-04792-f005:**
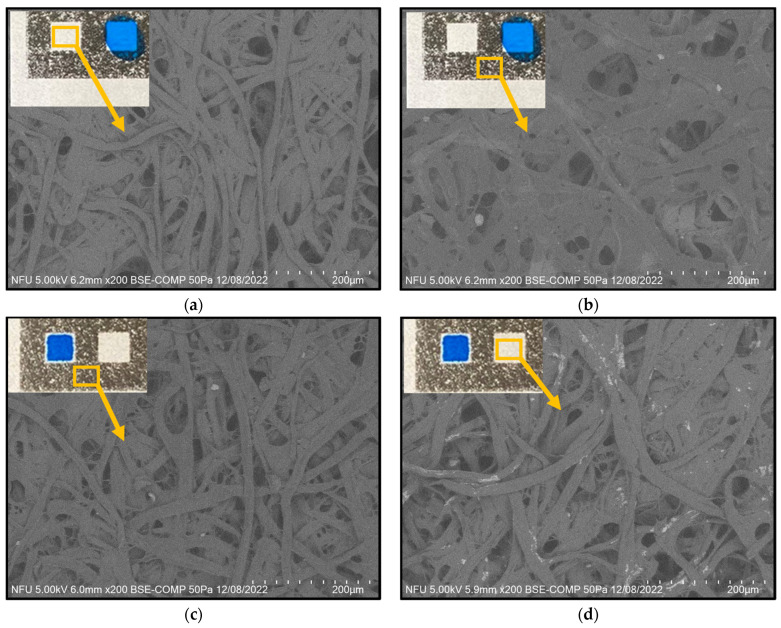
Hydrophobicity and SEM surface topography images of filter paper under different conditions: (**a**) before printing, (**b**) after printing, (**c**) after heating, and (**d**) after coating with NaCl and Tween 20.

**Figure 6 sensors-24-04792-f006:**
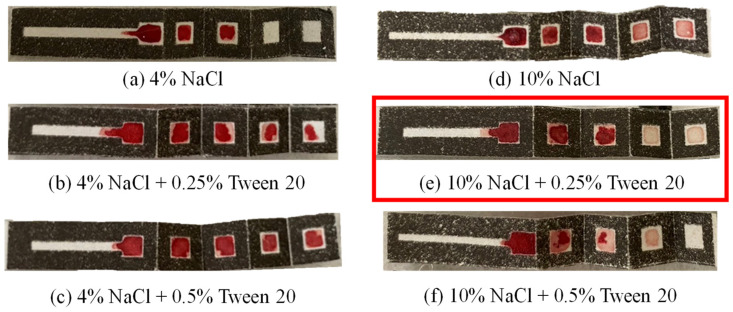
(**a**–**f**) Photographs of origami μPADs prepared with different concentrations of NaCl and Tween 20 following dropping of whole blood sample on reaction zone 1.

**Figure 7 sensors-24-04792-f007:**
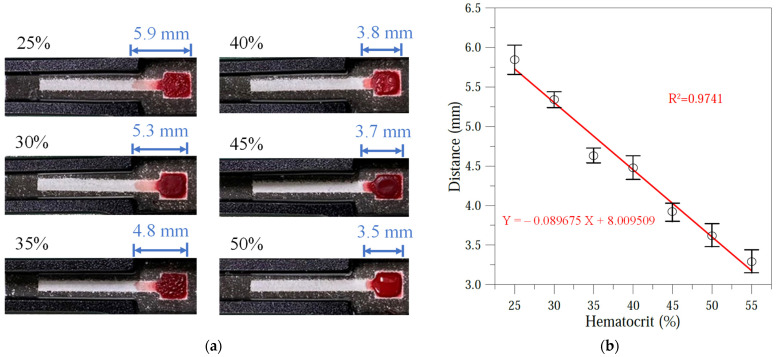
(**a**) Wicking distance of plasma on hematocrit test layer for different hematocrit values. (**b**) Calibration curve for variation in plasma flow distance with hematocrit value.

**Figure 8 sensors-24-04792-f008:**
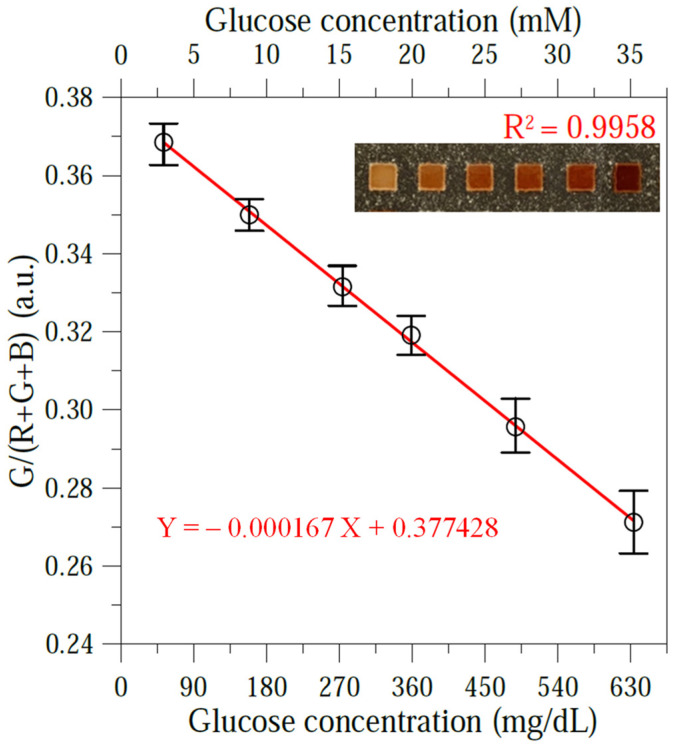
Calibration curve for G/(R + G + B) intensity measurements of glucose samples with different concentrations in range of 45–630 mg/dL. Inset images show glucose colorimetric assays of six control samples using KI color indicator.

**Figure 9 sensors-24-04792-f009:**
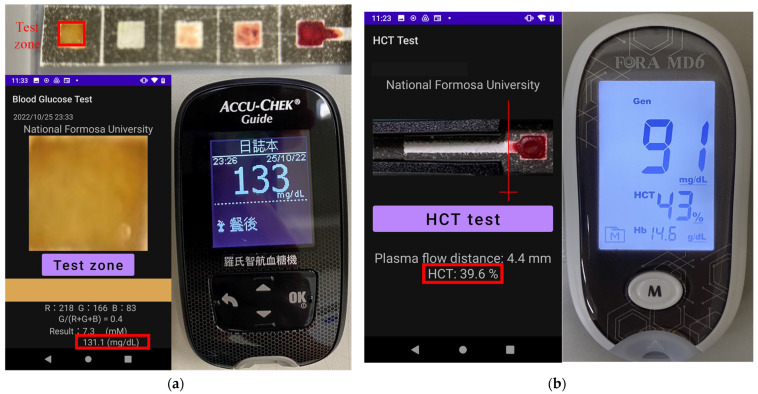
Comparison of measurement results obtained using origami μPAD with smartphone-based detection system and commercial meters: (**a**) glucose concentration and (**b**) hematocrit value.

**Table 1 sensors-24-04792-t001:** Comparison of glucose and hematocrit measurements obtained using developed origami μPAD and commercial meters for 13 real-world human whole blood samples.

	Blood Glucose (mg/dL)			Hematocrit (%)		
Samples	μPAD	Glucose Meter	Relative Error (%)	μPAD	Hematocrit Meter	Relative Error (%)
1	83.9	88	4.7	38.7	41	5.9
2	114.2	122	6.4	40.1	43	7.2
3	141.8	147	3.6	38.3	40	4.4
4	96.2	101	4.8	46.5	48	3.2
5	131.1	133	1.4	36.2	39	7.7
6	158.5	164	3.5	30.4	31	1.9
7	82.6	86	4.1	48.6	53	9.1
8	94.5	98	3.7	42.9	46	7.2
9	86.8	91	4.8	39.6	43	8.6
10	96.6	102	5.5	43.1	45	4.4
11	133.3	141	5.7	52.2	55	5.3
12	75.8	79	4.2	47.1	51	8.3
13	107.4	113	5.2	49.8	54	8.4

## Data Availability

The data presented in this study are contained within the article.

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
