# Peer review of "Paper-Based Microfluidic Analytical Device Patterned by Label Printer for Point-of-Care Blood Glucose and Hematocrit Detection Using 3D-Printed Smartphone Cassette"

_sensors, 2024, doi:10.3390/s24154792_

Round 1

Reviewer 1 Report

Comments and Suggestions for Authors

The authors describe a paper-based colorimetric device to measure hematocrit and glucose concentration in whole blood. The study is very interesting and well-designed, where the proposed sensor has been tested with 13 real human samples showing good agreement with commercial tests (error < 10%). There are only a few amended required before considering the paper for publication:

1. Have the authors used any blank to correct the RGB intensities for external light sources or spurious radiation that may affect measurements?

2. In Table 1, please insert the error range (min-max value) for each test.

Author Response

We're grateful for the reviewers' insightful comments and suggestions. We've carried out a 'point-to-point' revision of the manuscript, and our responses to each comment are provided in detail below.

Reviewer 2 Report

Comments and Suggestions for Authors

The article “Paper-based microfluidic analytical device patterned by label printer for point-of-care blood glucose and hematocrit detection using 3D-printed smartphone cassette” undoubtedly fits the topic of the journal and will be of interest to readers. However, I identified a number of significant comments:

1) What do the authors mean by "lines accuracy, efficiency, reliability"?

2) In lines 29-30, the paper mentions achieving an accuracy of at least 93.6% for blood glucose detection and 90.9% for hematocrit detection. However, it doesn't detail how these figures were calculated.

3) Figures 7a and 8. Accuracy do not assess  based on the linear range alone. To evaluate accuracy, a spike-recovery test should be conducted, along with studying the reproducibility of the analysis using at least 6 replicates per serum sample (expressed as CV,%) 

4) Figures 7a and 8 also do not specify the number of replicates used to construct the calibration curve. Additionally, it's unclear what the error bars represent (standard deviation, error of the mean?).

5) Given the method's sensitivity, the authors should specify the lower limit of detection and compare it with commercial assays and methods described in the literature.

6) Table 1 lacks details on how the error was calculated. Moreover, to demonstrate the new method's agreement with the reference standard, statistical methods such as Bland-Altman analysis should be employed.

General comment on the manuscript: The authors should enhance the validation section of the developed sensor. At the moment, it is impossible to say that an effective and accurate test has been developed. I recommend considering the manuscript for publication only after significant revisions and additional research.

Author Response

(The authors gave the same response as above.)

Reviewer 3 Report

Comments and Suggestions for Authors

The manuscript presents a paper-based microfluidic analytical device (µPAD) printed using a label printer, combined with a smartphone and a 3D-printed cassette to achieve quantitative detection of blood glucose and hematocrit. While the integration of label printing, 3D printing, colorimetric detection of glucose, and hematocrit testing based on the wicking distance of plasma is well-executed, I am afraid that none of these techniques are novel in isolation. Consequently, the manuscript might not offer new insights or methodologies from an academic perspective.

From an application standpoint, the manuscript emphasizes that the system is a low-cost yet accurate solution. However, commercially available medical-grade glucometers are priced as low as $10 on platforms like Alibaba. Therefore, it is recommended to provide a detailed cost analysis and comparison to substantiate the claimed cost-effectiveness of the proposed system.

Other points:

What is the response time for one testing?

In Figure 8, the linear fit between the RGB intensity ratio and glucose concentration is shown. What is the detection limit for the current system?

 The manuscript needs to elaborate on how consistent color detection is ensured in every test. Besides using standard light sources and fixed positions, how would using different brands and model of smartphones affect the accuracy of results?

 formatting error on line 380

Author Response

(The authors gave the same response as above.)

Round 2

Reviewer 2 Report

Comments and Suggestions for Authors

The revision is ok.  Authors significantly improved manuscript.

Reviewer 3 Report

Comments and Suggestions for Authors

The authors have responded all my concerns. 

Comments on the Quality of English Language

English language fine.